# Visual Colorimetric Detection of Edible Oil Freshness for Peroxides Based on Nanocellulose

**DOI:** 10.3390/foods12091896

**Published:** 2023-05-05

**Authors:** Xiongli Jiang, Jun Cheng, Fangwei Yang, Zhenyang Hu, Zhen Zheng, Yu Deng, Buyuan Cao, Yunfei Xie

**Affiliations:** 1State Key Laboratory of Food Science and Technology, Jiangnan University, No. 1800 Lihu Avenue, Wuxi 214122, China; 2School of Food Science and Technology, Jiangnan University, No. 1800 Lihu Avenue, Wuxi 214122, China; 3Collaborative Innovation Center of Food Safety and Quality Control in Jiangsu Province, Jiangnan University, No. 1800 Lihu Avenue, Wuxi 214122, China

**Keywords:** nanocellulose, hydrogels, edible oils, digital image colorimetry

## Abstract

Traditional methods for evaluating the edibility of lipids involve the use of organic reagents and complex operations, which limit their routine use. In this study, nanocellulose was prepared from bamboo, and a colorimetric reading strategy based on nanocellulose composite hydrogels was explored to monitor the freshness of edible oils. The hydrogels acted as carriers for peroxide dyes that changed color according to the freshness of the oil, and color information was digitized using UV-vis and RGB analysis. The sensitivity and accuracy of the hydrogel were verified using H_2_O_2_, which showed a linear relationship between absorbance and H_2_O_2_ content in the range of 0–0.5 and 0.5–11 mmol/kg with R^2^ of 0.9769 and 0.9899, respectively, while the chromatic parameter showed an exponential relationship with R^2^ of 0.9626. Surprisingly, the freshness of all seven edible oil samples was correctly identified by the hydrogel, with linear correlation coefficients greater than 0.95 in the UV-vis method and exponential correlation coefficients greater than 0.92 in the RGB method. Additionally, a peroxide value color card was established, with an accuracy rate of 91.67%. This functional hydrogel is expected to be used as a household-type oil freshness indicator to meet the needs of general consumers.

## 1. Introduction

Edible oils are a crucial component of our daily dietary intake, as they provide essential energy and fatty acids that are necessary for optimal physical development [1]. However, when stored, edible oils are highly susceptible to oxidation by light, heat, and oxygen, resulting in the production of peroxides, aldehydes, ketones, acids, and other small molecules that reduce the nutritional quality of the food [2]. Moreover, these oxidation by-products may be harmful to human health. Animal studies have shown that oxidized lipids may cause organ damage, inflammation, carcinogenesis, and advanced atherosclerosis [3]. Currently, there are several established methods for detecting lipid oxidation, such as the iodometric method, the thiobarbituric acid method, and the anisidine method. With technological advancements, fluorescence spectroscopy [4], high-performance liquid chromatography-mass spectrometry [5], near-infrared spectroscopy [6], infrared spectroscopy [7], nuclear magnetic resonance [8,9], and Raman spectroscopy [10,11] have also been utilized for lipid oxidation evaluation. However, these detection methods are limited by their expensive equipment and complex operation, which makes them unsuitable for daily use and impractical for the average consumer who wants to determine the edibility of their edible oils.

Colorimetry is a technique used to quantify the content of substances based on color shades and is divided into visual colorimetry and photoelectric colorimetry [12]. However, visual colorimetry is less accurate than photoelectric colorimetry because it is difficult to distinguish subtle color differences by observing the target color level with the naked eye [13]. Nowadays, the combination of colorimetric methods, image acquisition tools (such as cell phones, cameras, and scanners), and processing software (such as Adobe Photoshop [14], Image J [15], MATLAB [16], Pantone Studio [17], etc.) provides a new method for simple and accurate colorimetric detection, which can eliminate the subjective error of the observer and further improve the accuracy of visual observation in the field [18]. Digital image colorimetry has the advantages of simple operation, high speed, and low cost and is now used in the fields of medicine, chemistry, and food [19]. To promote standard specifications, color measurement systems are widely used for digital image colorimetry, and commonly used color models include RGB [20], HSV [21], and L*a*b* [22]. At present, digital image colorimetric methods are widely used for food safety testing of milk, fruits and vegetables, meat, etc. [23,24,25,26,27], but there are few reports on the testing of the freshness of edible oil. Additionally, these colorimetric indicators usually reflect the freshness of the food by sensing the change in pH of the system in the form of a film. However, changes in the environmental temperature and humidity during the use of the film-forming indicator may reduce their sensitivity and accuracy [28]. Therefore, to prevent false indications, it is recommended to use hydrogels as colorimetric indicators, as they have high hydrophilicity and can prevent changes in the environmental temperature and humidity [22,29].

Hydrogel is a three-dimensional cross-linked mesh material containing a large number of charged functional groups and hydrogen bonds after cross-linking. These properties allow hydrogels to absorb a large amount of water and provide optimal reaction conditions for liquid reaction systems [30]. Specific polymers can enhance the flexibility and volume of the hydrogel formed during the cross-linking process, making it easy to recycle in the liquid reaction system [31]. Due to its excellent characteristics, such as softness, flexibility, adjustable physical and chemical properties, porous structure, and excellent water absorption and retention properties [32], hydrogel has been widely used in agriculture [33], industry [34], biomedicine, and physiological health [35]. Although hydrogels have been studied as food freshness indicators, they have only been applied in the meat and seafood categories [15,28]. Nanocellulose is defined as cellulose with at least one dimensional size less than 100 nm [36]. It is a highly attractive material for preparing high-performance hydrogels due to its high aspect ratio, high crystallinity, high intermolecular binding energy, and unique surface chemistry [31]. Nanocellulose can be generally classified into cellulose nanofibers (CNF), cellulose nanocrystals (CNC), and bacterial cellulose (BC), depending on its size, morphology, and origin [37]. Nanocellulose has a wide range of applications in biological tissue engineering, drug carriers, medical materials, biological templates, wastewater treatment, composite materials, and other fields [38,39,40,41,42]. The 2,2,6,6-tetramethylpiperidine-1-oxyl radical (TEMPO) oxidation method can prepare nanocellulose that has more uniform size, more carboxyl groups on the surface, and reduces energy loss compared to mechanical and chemical separation methods [36].

Compared with wood fibers, bamboo is fast-growing, widely distributed, and has many processing by-products, which can greatly reduce production costs and resource consumption. Therefore, in this study, a highly peroxide-sensitive CNF-AM colorimetric hydrogel was prepared using bamboo as a raw material to evaluate the freshness of edible oil. Seven oil samples with high average daily usage were selected, and their freshness was quantitatively identified by combining UV-vis and color parameter analysis. The use of smartphones with RGB analysis and portable UV-vis meters made the method more cost-effective and portable.

## 2. Materials and Methods

### 2.1. Materials

Moso bamboo and edible oil were purchased from a local supermarket in Wuxi, Jiangsu. 2,2,6,6-tetramethylpiperidine-1-oxyl radical (TEMPO), sodium bromide (NaBr), sodium hypochlorite (NaClO), sodium hydroxide (NaOH) were obtained from Sinopharm Chemical Reagent Co., Ltd. (Shanghai, China). Sodium chlorite (NaClO_2_), anhydrous ethanol, acrylamide (AM), potassium persulfate (KPS), tetramethylenediamine (TEMED), ascorbic acid (AA), sulfosalicylic acid (SA), ferrous sulfate (FeSO_4_), glycerin, hydrogen peroxide (H_2_O_2_) were purchased from Aladdin Chemical Co., Ltd. (Shanghai, China). All chemicals were used as analytical reagents with no further purification.

### 2.2. Apparatus

The UV-vis absorption spectra were measured with a multifunctional microplate reader (800 TS, BioTek, Winooski, VT, USA). Photographs were collected by the camera (EOS R7, Canon, Tokyo, Japan), and the RGB of the images was analyzed through Image J (National Institutes of Health, Bethesda, MD, USA). Transmission electron microscopy (TEM) images were collected on an H-7650 microscope (HITACHI, Tokyo, Japan); the operating voltage was 120 kV. Scanning electron microscopy (SEM) was taken using the Quanta-200 (FEI, Hillsboro, OR, USA) with an accelerating voltage of 10 kV and a magnification of 150x. Fourier transform infrared (FTIR) spectra was recorded using FT-IR spectrometry (NEXUS, Thermo Nicolet Corporation, Madison, WI, USA). The X-ray diffraction (XRD) peaks were collected on an XRD-6000 diffractometer (Shimadzu, Kyoto, Japan). Zeta potential testing was conducted using a Malvern Mastersizer (ZS90X, Malvern Instruments Limited, Malvern, WOR, UK). Dynamic viscoelastic measurements of hydrogels were using a Discovery HR-3 Rheometer System (TA, New Castle, DE, USA).

### 2.3. Preparation of CNFs

CNFs were prepared according to a reported method with corresponding modifications [43]. To remove the lignin and hemicellulose, Moso bamboo was crushed to 100 mesh, soaked in 1 mol/L NaOH solution, and stirred continuously for 24 h. The bleached bamboo pulp was then placed in a solution of 5% (*w*/*v*) NaClO_2_ at 75 °C, continuously mechanically stirring and adjusting the pH to 5 with acetic acid and sodium acetate buffer solution. When the bamboo fibers turned white, the reaction was stopped, and the bleached bamboo pulp was rinsed with distilled water and stored in the refrigerator for later experimental research.

CNFs were prepared from the bleached bamboo pulp by TEMPO oxidation and subsequent mechanical treatment [36]. Briefly, the bleached bamboo pulp (5 g) was dispersed in deionized water (480 mL) under vigorous magnetic stirring. NaBr (0.5 g) and TEMPO (0.1 g) were added, followed by the drop-wise addition of NaClO solution (35 g) into the dispersion to initiate the TEMPO-mediated oxidation of cellulose. The pH of the dispersion was maintained at ten by the continuous addition of 1 M NaOH solution until NaOH was no longer consumed. The reaction was quenched with 2 mL of ethanol, and then the CNFs were washed thoroughly with distilled water and freeze-dried for 24 h. The surface carboxylate concentration of the CNFs was determined using an electrical conductivity titration method.

### 2.4. Preparation of CNF-AM Hydrogels

CNFs (0.05 g) were completely dissolved in deionized water (10 mL) with the aid of ultrasonic dispersion. Then, AM (0.01 g), KPS (0.005 g)m and TEMED (5 μL) were added successively under magnetic stirring. The CNF-AM gels were obtained by mixing for 10 min and incubating at room temperature for 3 h.

### 2.5. Preparation of Functional Hydrogels

FeSO_4_, AA, and SA were used as peroxide-sensitive indicators to impart color to CNF-AM gels. The indicator solution was prepared by mixing FeSO_4_, AA, and SA in a molar ratio of 1:10:20 in distilled water. The functional hydrogel was prepared by immersing the CNF-AM gels into the above indicator water solution for 12 h. Afterward, the residual indicator liquid on the surface of the gel was removed using deionized water.

### 2.6. Colorimetric Detection of Peroxide Values (PV) by Functional Hydrogels

The color difference of the functional hydrogels before and after exposure to the analyte was determined by a camera and a UV-vis spectrometer. Glycerol was used to simulate edible oil, and different concentrations of hydrogen peroxide were added to it. For the colorimetric signal, 100 μL of the glycerol solution with varying peroxide values (0, 0.031, 0.062, 0.125, 0.250, 0.500, 1, 3, 5, 7, 9, 11 mmol/kg) were added to a 250 μL functional hydrogel and reacted for twenty minutes. For UV-vis spectroscopy, functional hydrogels were transferred into 96-well plates, measured at a maximum absorption wavelength of 510 nm, and the measured values (ΔA) were recorded. The photographs of the functional hydrogels were taken with a camera. The RGB (red, green, and blue) analysis of the photographs was performed using Image J, and the color difference value (ΔR) was calculated using the following equation:(1)$$\Delta R=\sqrt{{\left(R-R_{0}\right)}^{2}+{\left(G-G_{0}\right)}^{2}+{\left(B-B_{0}\right)}^{2}}$$
where *R*, *G*, *B* are the redness, greenness, and blueness of the indicator hydrogels, respectively; *R*_0_, *G*_0_, *B*_0_ are the values of the control hydrogels.

### 2.7. Oxidation of Edible Oil

The accelerated oxidation of monounsaturated oils (camellia oil, canola oil, olive oil, and peanut oil) and polyunsaturated oils (corn oil, soybean oil, and linseed oil) was performed using the Schall oven test [2]. The edible oil was placed in glass bottles with caps and kept in an oven at a temperature of 62 ± 1 °C. The bottles were shaken every 12 h, and their positions were randomly changed in the oven. To obtain edible oils with varying degrees of oxidation, samples were taken at regular intervals and quickly stored in a −20 °C refrigerator. The peroxide values (PV) of edible oil were measured according to the National Food Safety Standards of China (GB5009.227–2016) to evaluate the oxidation degree of oil [2].

### 2.8. Application of Functional Hydrogel in Oil Spoilage

To determine the degree of oil spoilage, an equal volume of deionized water was added to the oxidized oil sample in a centrifuge tube and ultrasonically mixed to form an emulsion. The blank control was prepared by not adding any oil sample. Next, 100 μL of the above mixture was added to the functional hydrogels, and the color difference (ΔR) and the UV-vis absorption spectra (ΔA) were measured after twenty minutes.

## 3. Results and Discussion

### 3.1. Characterization of Cellulose and CNFs

Bamboo raw materials contain cellulose, hemicellulose, and lignin. Appendix A shows pictures of bamboo raw materials in various treatment stages. After delignification and bleaching treatment, large-size cellulose microfibrils were obtained, which were subsequently used to prepare carboxylated nanocellulose fibrils (CNFs) through TEMPO oxidation, introducing negative charges on the surface of CNFs [28,44]. The oxidation degree and surface charge density of CNFs were determined to be 12.96% and 0.8 mmol/g, respectively, which promoted nanocellulose stripping and improved water dispersion. The surface of CNFs was abundant with carboxyl groups, which facilitated the creation of CNF-AM hydrogels since carboxyl groups can form hydrogen bonds with amino groups.

The raw materials of bamboo consist of cellulose, hemicellulose, and lignin (as shown in Appendix A). After delignification and bleaching treatment, large cellulose microfibrils were obtained, which were subsequently used to prepare carboxylated nanocellulose fibrils (CNFs) through TEMPO oxidation, introducing negative charges on the surface of CNFs [28,44]. The oxidation degree and surface charge density of CNFs were determined to be 12.96% and 0.8 mmol/g, respectively, which promoted nanocellulose stripping and improved water dispersion. The surface of CNFs was abundant with carboxyl groups, which facilitated the creation of CNF-AM hydrogels since carboxyl groups can form hydrogen bonds with amino groups.

TEM analysis showed that the dimensions of CNFs were smaller than those of cellulose (Appendix A). The width of nanocellulose was less than 100 nm, and the average length of CNFs was analyzed in Appendix A; the CNFs’ lengths were mainly distributed between 200~400 nm. As seen in the zeta potential results in Appendix A, both cellulose and CNFs were rich in negative charge. However, the surface potential of CNFs is greater than that of cellulose, indicating that the surface of CNFs has been successfully carboxylated. These results demonstrated the successful preparation of surface carboxyl-rich nanocellulose from bamboo, providing a foundation for the subsequent hydrogel preparation.

### 3.2. Cross-Linking of CNF-AM Hydrogels

Double cross-linking of CNF-AM hydrogels can address the limitations of traditional nanocellulose hydrogels, such as poor flexibility and single functionality, and significantly enhance their physical and chemical properties [45]. Both CNF hydrogels and CNF-AM hydrogels exhibit smooth shear-thinning rheological curves, but the latter had lower mobility and higher viscosity than the former (Appendix A). This could be due to the addition of acrylamide, which produces composite hydrogels with double network structures of rigid and flexible polymer interpenetration, thereby improving the mechanical properties of the hydrogels. The storage modulus G’ of the hydrogels was greater than the loss modulus G”, indicating that the hydrogels were stable and behaved as viscoelastic hydrogels (Appendix A).

### 3.3. Morphological Characterization of CNF-AM Hydrogels

The X-ray diffraction (XRD) pattern of cellulose, CNF, and CNF-AM gels is shown in Figure 1a. All the XRD patterns exhibited a typical cellulose I crystalline structure, with XRD peaks located at 15.1°, 16.2°, and 22.1° corresponding to ($\bar{110}$), (110), and (200) reflection planes, respectively [46]. The X-ray diffractogram indicated that the TEMPO treatment and the addition of AM did not influence the crystalline structure of cellulose. Additionally, the crystallinity of cellulose, CNF, and CNF-AM gels was calculated to be 44.9%, 36.7%, and 31.6%, respectively, based on the height of the peaks in the XRD spectra. The higher crystallinity of cellulose could be due to the pretreatment to remove the non-cellulose components, while the decrease in crystallinity of CNF could be attributed to the ultrasonic crushing process, which disrupts the hydrogen bonds and crystalline regions, resulting in a reduction in crystallinity. After the formation of the CNF-AM hydrogel, the crystallinity was further reduced, and the XRD pattern diffraction peaks decreased in height, broadened in width, and flattened in shape. The altered pattern suggests that the addition of AM may lead to the formation of stronger intermolecular forces, which contributes to good biocompatibility and promotes the formation of a more regular and compact gel structure.

The molecular vibration information of cellulose, CNF, and CNF-AM gels was observed from the FT-IR spectra (Figure 1b). The bands at 3331 cm^−1^ and 2879 cm^−1^ corresponded to hydrogen bonding of OH…O and vibration of –OH, as well as C–H stretching vibrations of –CH_2_ and -CH_3_ in all cellulose samples, respectively [44,47]. The absorption peaks at 1159 cm^−1^ and 878 cm^−1^ were characteristic frequency peaks of the pyran ring in cellulose, which were caused by asymmetric stretching vibration absorption peaks of C–O–C and skeletal vibrations of –C–H, respectively [44,45]. These bands showed no significant change in position or peak intensity, indicating that the application of AM did not alter the structure of the cellulose chain. Meanwhile, the inverted peak at 2360 cm^−1^ indicated the presence of water in the three samples, which was difficult to remove completely due to strong hydrogen bonding interactions between cellulose and water, despite all samples being freeze-dried [48]. In contrast, a strong new peak appears at 1599 cm^−1^ in the spectra of CNF and CNF-AM gels due to the oxidation of hydroxyl groups to carboxyl groups after TEMPO oxidation [49]. From the FT-IR spectra of AM and CNF-AM gels, a strong band at 1668 cm^−1^ corresponds to the stretching vibrations of C=O, but it shifts to 1651 cm^−1^ due to the formation of hydrogen bonds. Moreover, no new vibrational absorption peaks appear in the CNF-AM gels IR spectrum, indicating only the physical blending of the two [45].

SEM was used to characterize the appearance morphology of CNF (Figure 1c) and CNF-AM gels (Figure 1d); it can be seen that the CNF and CNF-AM gels were formed by three-dimensional structures consisting of a large number of connected irregular sheets, which resulted from the self-assembly of rod-like nanocellulose into thin films by hydrogen bonding interactions after freeze-drying [47]. Additionally, while the surface of CNF gels was smooth and flat, the surface of CNF-AM gels was rough and had obvious protrusions, indicating that CNF and AM have been linked together through hydrogen bonds. CNF was mostly embedded in the pore walls, which also increased the specific surface area of CNF-AM gels [50]. The rough surfaces of these hydrogels may help increase the contact surface of the reaction [51]

### 3.4. Formation Mechanism and Properties of CNF-AM Hydrogels

Based on structural characterization and microstructure analysis, it can be concluded that the gelation phenomenon of CNF-AM hydrogels was mainly due to hydrogen bonding, resulting in a double network gel structure. The presumed mechanism for preparing CNF-AM hydrogel is shown in Figure 2. Firstly, AM is transformed into polyacrylamide (PAM) through the action of KPS and TEMED. KPS acts as the initiator of the grafting reaction, while TEMED acts as the cross-linking promoter. The surface of the CNF, prepared by TEMPO oxidation, was rich in carboxyl and hydroxyl groups, which can form hydrogen bonds with carbon and amino groups on PAM. Additionally, CNF and AM can also form hydrogen bonds spontaneously, respectively, and finally form a stable three-dimensional network structure. The CNF-AM hydrogel is a physically cross-linked hydrogel, which is known to have better adsorption than chemically cross-linked hydrogels [52]. Moreover, the high specific surface area of nanocellulose can increase the adsorption of peroxide dyes, and the small molecules (ions) inside the hydrogel can diffuse freely in its three-dimensional network. During the vegetable oil pretreatment process, the mixed emulsion of water and oil is formed by ultrasonic. It is assumed that the rich pore structure and good mechanical properties of the double network hydrogel can be used to detect the freshness of edible oil.

### 3.5. Response of Functional Hydrogels to Peroxide

In order to study the response of functional hydrogels to peroxide in edible oil, glycerol and hydrogen peroxide were used to simulate oil samples with different oxidation levels. A blank control was used to exclude environmental impacts. As shown in Appendix A, the pure hydrogel did not absorb in the range of 410–560 nm, but after the addition of hydrogen peroxide, the maximum absorption peak was observed at 510 nm. The functional hydrogels changed color from colorless to deep red as the concentration of hydrogen peroxide increased from 0 to 11 mmol/kg. The relationship between ΔA and hydrogen peroxide concentration was found to be exponential, with a correlation coefficient of 0.9633 (Figure 3a). In addition, the ΔA had linearly enhanced in the low and high concentration ranges, respectively. At low concentrations (range 0~0.5 mmol/kg), the R^2^ was 0.9769 (Figure 3b), and at high concentrations (range 0.5~11 mmol/kg), the R^2^ was 0.9899 (Figure 3c). The experimental results indicated that the UV-vis spectrophotometric method can detect peroxides effectively.

To obtain the quantitative color information, a camera was used to take the photographs, and Image J software was employed to divide the original photographs into three channels, including red, green, and blue. The physical image of the functional hydrogel reacting with hydrogen peroxide showed that the color becomes darker with increasing hydrogen peroxide concentration (Figure 3d). As expected, the ΔR enhanced linearly in the low-concentration range and tended to be flat in the high-concentration range (Figure 3e). The R^2^ was 0.9445 over the range of 0~0.5 mmol/kg of hydrogen peroxide concentration (Figure 3f). Meanwhile, the overall hydrogen peroxide concentration was found to be exponentially related to ΔR with an R^2^ of 0.9626. This response effectiveness verified that the colorimetric method could be applied to monitoring edible oil spoilage. However, the visual determination limit of the functional hydrogels prepared in this study should not be considered as the visual determination limit of other peroxides due to the stronger reactivity of H_2_O_2_ than long-chain peroxides. In addition, unlike other peroxide detection methods, it is a suitable detection limit that can accurately reflect the oxidation status of edible oils rather than a very low detection limit.

### 3.6. Application of Functional Hydrogel on the Edible Oil

To assess the practicality of using this functional hydrogel to determine the freshness of edible oils, we selected seven commonly used edible oils for household cooking, including monounsaturated oils (camellia oil, canola oil, olive oil, and peanut oil) and polyunsaturated oils (corn oil, soybean oil, and linseed oil). We then obtained oil samples with varying degrees of freshness using the Schall oven oxidation method and tested them using the functional hydrogel through colorimetric assays. To validate the accuracy and reliability of our proposed method, we also used the authoritative standard method, Iodometric titration (GB5009.227—2016), to measure the concentration of peroxide during the oxidation of the oils.

During the accelerated oxidation of edible oils, hydroperoxides are formed through the cleavage of unsaturated fatty acids. As shown in Appendix A, the peroxide values of camellia oil, canola oil, olive oil, peanut oil, corn oil, and soybean oil increased during the oxidation process from 0 to 15 days, while the peroxide values of linseed oil increased during the oxidation process from 0 to 72 h. The results of the iodometric titration method were in close agreement with the ΔA value changes of the functional hydrogel for both monounsaturated and polyunsaturated oils. This was further confirmed by the linear relationship between PV and the ΔA value. Notably, the linear correlation coefficients for all samples were above 0.95 (Figure 4).

According to the Chinese national standard GB 2716–2018, the PV limit for edible oils is 9.82 mmol/kg. Due to the different fatty acid compositions of various oil samples, the time required for PV to reach the limit value also varies. Obviously, linseed oil is the most susceptible to oxidation, with its PV exceeding the limit value of the national standard after about 25 h of oxidation, rendering the oil inedible. Hence, we calculated the edible limit ΔA values by fitting equations to different oil samples, which can be used to determine the edibility of the oil. The calculated edible limit ΔA values were 2.21, 1.47, 1.39, 1.09, 2.25, 2.02, and 1.49 for camellia oil, olive oil, canola oil, peanut oil, corn oil, soybean oil, and linseed oil, respectively. These results indicate that the functional hydrogel can accurately predict the freshness of edible oils and has broad applicability. The relationship between the ΔR signal and PV is shown in Figure 5, which is similar to the simulated results. Both monounsaturated and polyunsaturated oils follow an exponential function, indicating that the image analysis method can also be applied to analyze the freshness of actual oil samples.

### 3.7. Accuracy of Standard Colorimetric Card

A standard colorimetric card was created based on the color signal obtained from RGB analysis, as shown in Figure 6. To verify the accuracy of the established standard colorimetric cards, a selection of oils, including blending oil, sunflower oil, rice oil, linseed oil, soybean oil, and canola oil, were chosen for experimentation. Appendix A shows the physical diagram of the peroxide-responsive hydrogel after reacting with different oil samples, while Appendix A displays the corresponding PV of each oil sample. The peroxide content of the oil samples can be visualized by the color difference, with darker colors indicating higher peroxide values. The results of the GB 5006.229-2016 standard test showed that 17 samples had exceeded the national standard limit value and were not recommended for consumption, while 31 samples did not exceed the limit value and were deemed safe for consumption. As shown in the confusion matrix in Figure 7, among the 31 samples that were actually edible, 27 were correctly identified as such, while four were misclassified as inedible. Among the 17 samples that were actually inedible, all were correctly identified as such, resulting in an accuracy rate of 91.67% for the method. It is speculated that the reason for the lower detection accuracy and precision of the colorimetric method compared to the UV-vis method and digital image method is that the color difference between oil samples with peroxide values near the limit value of the national standard is small, leading to misclassification of oil samples near the limit value as sour oil samples based on simple visual judgment.

Compared to previous studies on food freshness indicators, this study’s hydrogel indicator has several advantages. Most food freshness indicators are in the form of films fixed on the surface of food packaging, which can be affected by changes in storage conditions that reduce sensitivity and accuracy. Previous studies on colorimetric hydrogels have mainly focused on monitoring meat and seafood, with relatively few reports on detecting oxidative deterioration of edible oils (Table 1). In this study, we developed an edible oil freshness indicator suitable for home use that exhibits a color response related to peroxide value and can be applied to seven commonly used edible oils. This method is highly versatile and environmentally friendly.

## 4. Conclusions

In summary, a peroxide colorimetric hydrogel was prepared to evaluate the oxidation of edible oils. The color intensity of the hydrogel increased with the peroxide value of the oil. The colorimetric information of the hydrogel was obtained by UV-vis and RGB analysis, both of which were sensitive to H_2_O_2_ in the range of 0–0.5 mmol/kg. Notably, the colorimetric hydrogel correctly assessed the freshness of all seven oil samples, and a standard colorimetric card was constructed based on the colorimetric results, with peroxide values ranging from 0–12 mmol/kg and a validation accuracy of 91.67%. Therefore, the proposed method is expected to be a practical tool for evaluating the oxidation status of edible oils and has potential applications in food freshness testing. In the future, a portable detection device can be designed, and corresponding software can be developed in conjunction with smartphones to enhance the method’s accuracy and promote its daily use.

## Figures and Tables

**Figure 1 foods-12-01896-f001:**
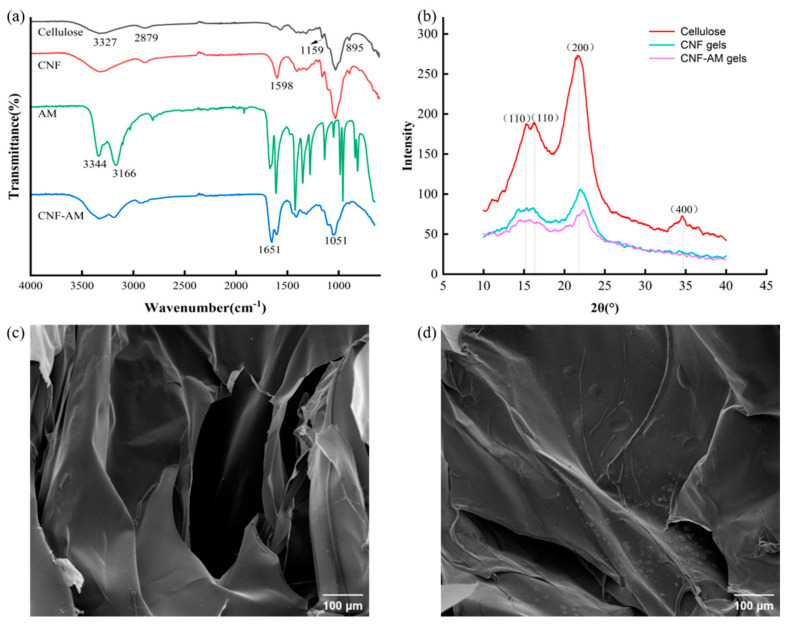
(**a**) FT−IR spectra of cellulose, CNF gels, AM and CNF−AM gels, (**b**) X-ray diffraction of cellulose, CNF and CNF−AM gels, SEM image of (**c**) CNF gels, (**d**) CNF−AM gels.

**Figure 2 foods-12-01896-f002:**
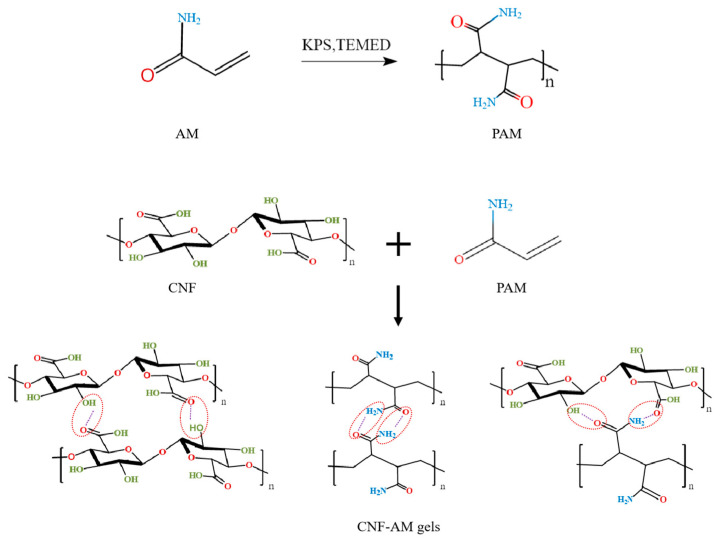
Possible hydrogen bonds in CNF-AM hydrogel.

**Figure 3 foods-12-01896-f003:**
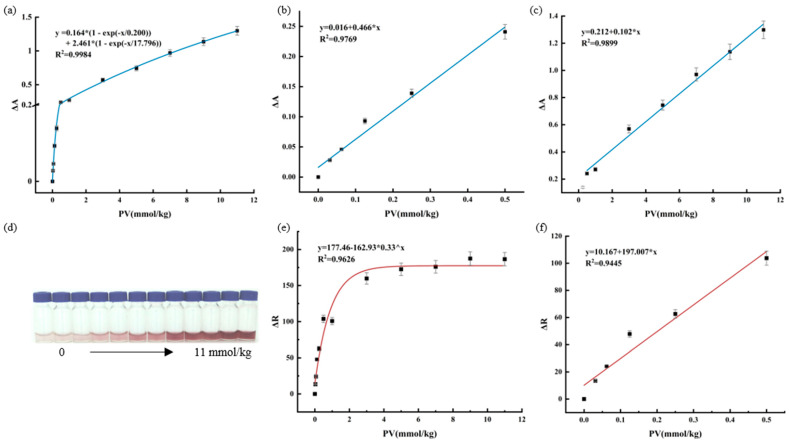
Relationship between the ΔA and H_2_O_2_ concentration of functional hydrogels. (**a**) PV range 0−11 mmol/kg, (**b**) PV range 0−0.5 mmol/kg, (**c**) PV range 0.5−11 mmol/kg. Relationship between the ΔR and H_2_O_2_ concentration of functional hydrogels. (**d**) An image of functional hydrogels obtained with a camera, (**e**) PV range 0−11 mmol/kg, (**f**) PV range 0−0.5 mmol/kg.

**Figure 4 foods-12-01896-f004:**
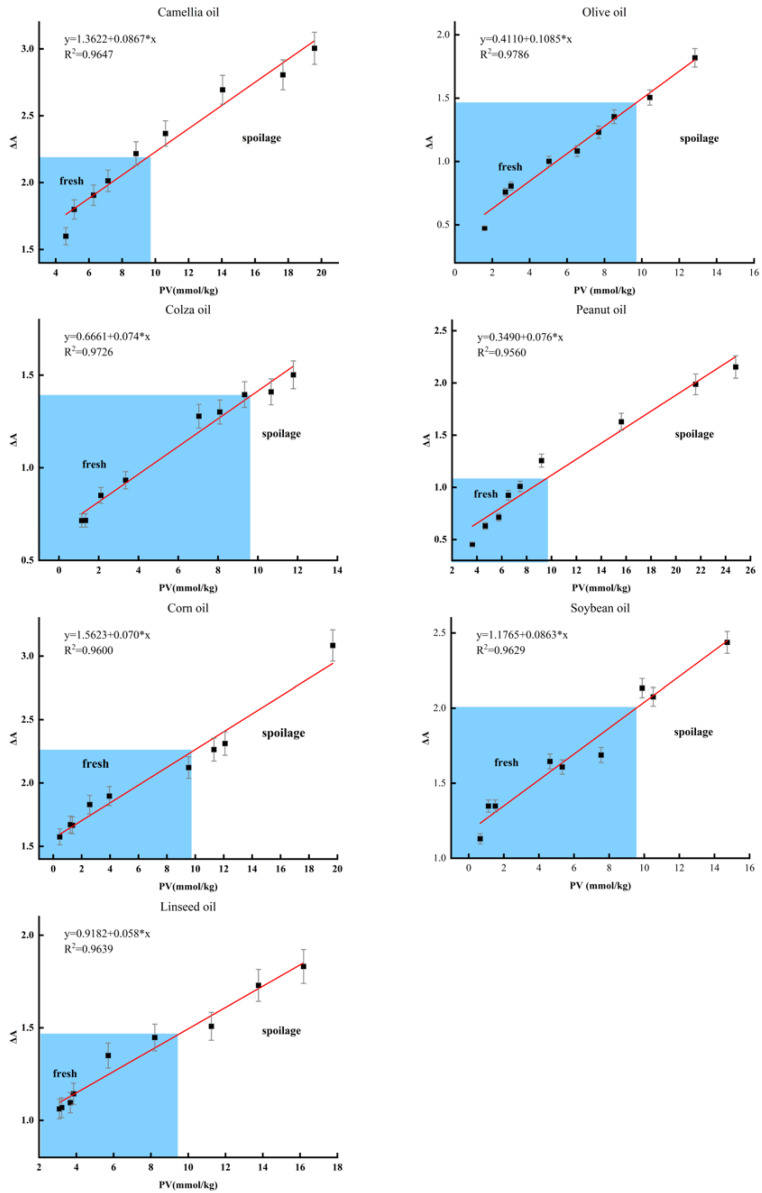
Correlation of the results between the PV and ΔA in monitoring the freshness of edible oil.

**Figure 5 foods-12-01896-f005:**
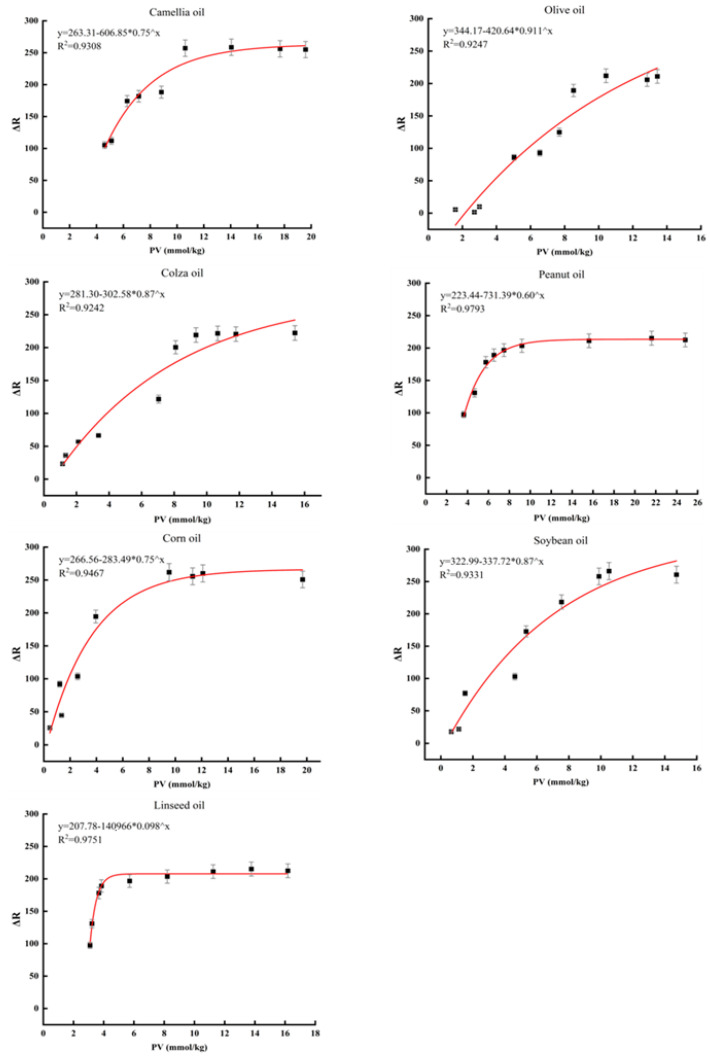
Correlation of the results between the PV and ΔR in monitoring the freshness of edible oil.

**Figure 6 foods-12-01896-f006:**
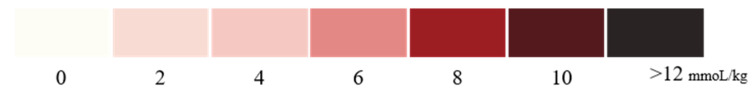
Standard colorimetric card for PV.

**Figure 7 foods-12-01896-f007:**
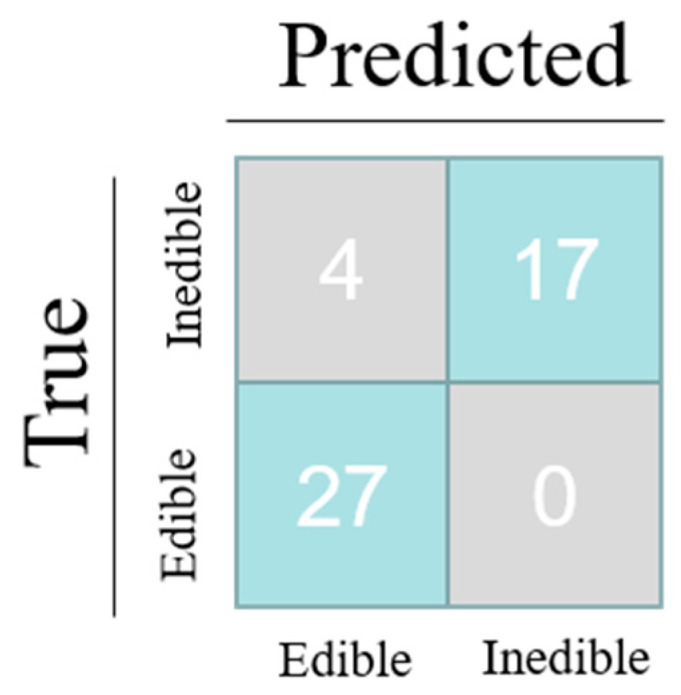
Confusion matrix.

**Table 1 foods-12-01896-t001:** Summary of the application of colorimetric hydrogel and colorimetric detection of edible oil.

Materials	Indicators	Detected Substances	Mechanism	Application	References
Chitin nanowhiskers film	Congo red and hydroxylamine sulfate	Aldehydes	Hydroxylamine sulfate reacts with aldehydes, releasing sulfuric acid and leading to a decrease in pH	Sunflower oil	[2]
Polyethyleneimine (PEI) Solution	PEI	2-tert-butyl-1,4-benzoquinone (TBBQ)	PEI link with TBBQ through Michael addition to form colored adducts	TBBQ-spiked soybean oil and peanut oil	[53]
polyvinyl alcohol film	Schiff’s reagen	Aldehydes	Schiff’s reagent and aldehydes form colored compounds	Sunflower oil	[54]
Agarose hydrogel	Silver-doped Prussian blue nanoparticles	Volatile amines	Decomposition of silver-doped Prussian blue nanoparticles caused by volatile aldehydes	Shrimp and fish	[15]
Sugarcane bagasse nanocellulose hydrogel	Bromothymol blue and methyl red	CO_2_	CO_2_ levels increase with the spoilage of chicken, leading to a decrease in pH	Chicken	[28]
N,N-dimethyl acrylamide-co-methacryloyl sulfadimethoxine hydrogel	N,N-dimethyl acrylamide, methacryloyl sulfadimethoxine	-	Changes in pH lead to changes in the transparency of the hydrogel	-	[55]
Agarose hydrogel	β-D-glucose pentaacetate (β-D-GP) and silver ions	Biogenic amines	Biogenic amines hydrolyze β-D-GP to β-D-glucose, and β-D-glucose reduces silver ions to silver nanoparticles with a color change.	Fish	[56]
Nanocellulose hydrogel	Anthocyanins	Total volatile basic nitrogen (TVB-N)	TVB-N causes a change in pH with a color change	Pork	[29]
Alginate-methylcellulose blend hydrogel	Bromothymol blue	TVB-N	TVB-N causes a change in pH with a color change	A minced pork	[22]
CNF-AM hydrogel	Fe(II), SA, and AA	Peroxides	Peroxides oxidize Fe(II) to Fe(Ⅲ) with a color change	Seven types of edible oils	In this study

## Data Availability

All related data and methods are presented in this paper. Additional inquiries should be addressed to the corresponding author.

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
