# Peer review of "Visual Colorimetric Detection of Edible Oil Freshness for Peroxides Based on Nanocellulose"

_foods, 2023, doi:10.3390/foods12091896_

Round 1
Reviewer 1 Report
1) in introduction section clarify the originalty and the objective of this study
2) some questions were adressed in the authors:
- What is the main question addressed by the research?
- Do you consider the topic original or relevant in the field? Does it address a specific gap in the field?
- What does it add to the subject area compared with other published material?
Author Response
Dear Reviewer:
We sincerely thank you for the valuable feedback that we have used to improve the quality of our manuscript. Our response is shown in the attachment.

Reviewer 2 Report
The paper started as an interesting paper to read but starting with results the paper become hard to be read due to the abundance of information, and due to the loose of the declared aim of paper which is related to the quantitatively identification of the oxidation degree (their freshness) of 7 edible monounsaturated oils (camellia oil, canola oil, olive oil, and peanut oil) and polyunsaturated oils (corn oil, soybean oil, and linseed oil) by combining UV-VIS and color parameter analysis. Surprising, a (too) large part of the paper deals with advanced characterization of cellulose, cellulose nanofibers (CNFs), cellulose nanofibers – acrylamide hydrogels using a lot of methods and a lot of details. Generally this could be fine, but this intention was not declared in the paper’ title, abstract or introduction, which means in the aim of the paper. In the same time, many results really related to the declared aim of the paper are presented in the supplementary materials. I did not found useful the TEM, SEM, viscosity, shear modulus, zeta potential, etc (I din not find any relevance for all of these in the context of Visual colorimetric detection of edible oil, the correlation is missing!). Moreover, if the Abstract and introduction is well written then the paper becomes sloppy and unfinished. There are many mentions of “Error! Reference source not found” and many yellow highlighted characters appears. It looks like you submitted a working version and not the final document (did you not read and approved the submission?). Due to this issues my recommendation to the editor is that the paper to be rejected and resubmitted with the necessary corrections. In my opinion you should decide what you would like to submit: i) the “Visual colorimetric detection of edible oil freshness for peroxides based on nanocellulose (here please verify the English)” or ii) the preparation and characterization of CNFs and CNF-AM hydrogels for visual colorimetric detection of edible oil freshness? If is the second one then you should rewrite completely the abstract and introduction and to declare the aim of the paper reflecting that. If is the first version then you should bring important information from supplementary materials and you should place the characterization to the supplementary material. Or you may decide to write not one but two papers. But in the present form is not readable. Other aspects. 1. Page 3, section 2.3: Please explain “dramatical magnetic stirring”. 2. Equation 1. In can’t differentiate between for example (and simplicity) i) R(100), G(150), B(200) and R0(100), G0(150), B0(220) and ii) R(100), G(150), B(200) and R0(120), G0(150), B0(200). The differences appear in the first case in the blue region and in the second case in the red region (of the VIS spectra). You may consider analyzing the light on all three channels. 3. Figures 5 and 6, and S3 and S4 can be represented only on two columns and then enlarged. Many features are not visible. 4. The references to the subfigures from Fig. 5 are not mentioned in the proper way.
Author Response

(The authors gave the same response as above.)

Reviewer 3 Report
The scientific content of the ms. is referring to the development of a visual colorimetric detection method of edible oil freshness for peroxides based on nanocellulose. The present study is well-written and a detailed effort is described. In addition, the eligibility of the method for practical application is evident. To my opinion minor revision of this ms. is required concerning the content and thus some points should be examined/ reconsidered by the authors in order to this article be accepted.
General comments on this experimental work – which supports my proposal for processing after minor revision – and revision points/comments/suggestions to be taken into account:
(a) General comment: A sustainable effort to improve the English language used should be performed. Many syntax and grammar errors exist in the ms. and phrases that could be presented more clearly. For instance, “proper bodily development”, “a peroxide colorimetric hydrogel was prepared to evaluate the oxidation of edible oils, with the color deepening with increasing peroxide value”.
(a) Introduction: “At present, the digital image colorimetric method is widely used for food safety testing of milk, fruits and vegetables, meat, etc., but there are few reports on the testing of the freshness of edible oil.” Add representative publications for this statement.
(b) Results and discussion: Many references are not cited properly in the ms. They are indicated as “Error! Reference source not found” instead of the right number of the citation. In addition many highlighted fonts remain in the text. The authors should be careful with the format.
Author Response

(The authors gave the same response as above.)

Round 2
Reviewer 2 Report
Thank you for implementing my suggestions.